# Indoxyl Sulfate and Autism Spectrum Disorder: A Literature Review

**DOI:** 10.3390/ijms252312973

**Published:** 2024-12-03

**Authors:** Zoë R. Hill, Christina K. Flynn, James B. Adams

**Affiliations:** 1Barrett, The Honors College, Arizona State University, Tempe, AZ 85287, USA; zhill4@asu.edu; 2Biodesign Center for Health Through Microbiomes, Arizona State University, Tempe, AZ 85287, USA; ckflynn@asu.edu

**Keywords:** autism, indoxyl sulfate, metabolite

## Abstract

Indoxyl sulfate—a bacterially derived metabolite—has been identified as a toxin that is elevated in children with autism spectrum disorder (ASD). As a neurotoxin, uremic toxin, nephrotoxin, cardiotoxin, osteotoxin, and myotoxin, indoxyl sulfate has been associated with several other conditions, including chronic kidney disease, acute kidney injury, Parkinson’s disease, cognitive disorders, and mood disorders such as anxiety and depression. Indoxyl sulfate is derived from bacterial modification of host tryptophan, and elevated levels of indoxyl sulfate are associated with decreased levels of important neurotransmitters including serotonin, dopamine, and norepinephrine. This article will review what is currently known about indoxyl sulfate in relation to ASD and its comorbidities. A systematic review identified six studies of levels of indoxyl sulfate in children with ASD. All six studies found that indoxyl sulfate was significantly elevated in the urine of children with ASD compared to typically developing children. Through this review, indoxyl sulfate was identified as a toxic microbially derived metabolite that is significantly increased in a subset of children with ASD and may contribute to both core and co-morbid ASD symptoms.

## 1. Introduction

Autism spectrum disorder (ASD) is a complex neurodevelopmental disorder characterized by impaired social functioning and communication, repetitive behavioral patterns, and specialized interests [1,2]. ASD is estimated to occur in 1 in 36 children ages eight and under in the United States, with the disorder occurring in boys at a rate four times higher than in girls [2]. Children with ASD can experience a variety of cognitive, social, behavioral, and health symptoms, ranging from low- to high-functioning. The cause of ASD and its symptoms is presently unknown.

ASD is currently diagnosed through parental interviews, physician observation, and standardized behavioral scales measured by observing the patient in a set of social and behavioral tasks [1]. The variability of symptoms falling under the definition of ASD makes the disorder difficult to diagnose and may delay effective medical intervention and behavioral therapy for the child. Diagnosis at older ages has been shown to contribute to the affected child developing worse social and behavioral outcomes and may require a greater number of therapies to stabilize [1]. To date, there is no reliable physiological biomarker for ASD. Understanding the causes of autism is essential for the creation of more effective methods of diagnosis and treatment.

Evidence linking the gut microbiome and gastrointestinal dysfunction to ASD has increased in recent years [1,2,3,4,5]. Children with ASD are over four times more likely to experience symptoms of gastrointestinal dysfunction such as abdominal pain, diarrhea, constipation, bloating, and intestinal inflammation [2]. These symptoms could be evidence of abnormal gut microbiome compositions and the presence of potentially toxic microbial metabolites.

One microbially derived metabolite of high interest in ASD is 3-indoxyl sulfate, simply known as indoxyl sulfate (Figure 1). Indoxyl sulfate is an aryl sulfate co-produced by bacteria in the gut and the host’s metabolism through the tryptophan metabolic pathway [6]. Indoxyl sulfate has been classified as a neurotoxin, uremic toxin, nephrotoxin, cardiotoxin, osteotoxin, and myotoxin [6]. High levels of indoxyl sulfate are associated with several conditions, including chronic kidney disease, acute kidney injury, Parkinson’s disease, and various mood disorders [7,8,9]. Additionally, indoxyl sulfate has been discovered in abnormal concentrations in children with ASD [1]. Elevated levels of a toxin such as indoxyl sulfate may contribute to a variety of symptoms pertaining to ASD, making the metabolite particularly relevant when researching potential biomarkers for the disorder.

This paper will investigate the potential impact of indoxyl sulfate on ASD, beginning with the metabolic genesis of indoxyl sulfate and how it is processed by the human body. Then, the various toxin classifications of indoxyl sulfate will be explored, including its classification as a neurotoxin, uremic toxin, nephrotoxin, cardiotoxin, osteotoxin, and myotoxin. Further, other disorders besides ASD with evidence of abnormal indoxyl sulfate levels will be examined, with particular attention paid to comorbidities of ASD. Evidence pertaining to indoxyl sulfate’s influence in symptoms of ASD will be reviewed, and potential treatments for abnormal indoxyl sulfate concentrations will be identified. Finally, recommendations for future research will be suggested.

## 2. Methods

A systematic literature review was conducted to find all studies involving autism and indoxyl sulfate. The literature review was conducted using PubMed, Google Scholar, and the Arizona State University Library research database. To find relevant articles, key words related to the topic were searched, including “indoxyl sulfate”, “uremic toxin”, “Autism”, “indican”, “indole”, “tryptophan metabolism”, “3-indoxyl sulfate”, “indoxyl sulfate uremic toxin”, “indoxyl sulfate cardiotoxin”, “indoxyl sulfate osteotoxin”, and “indoxyl sulfate myotoxin”. A total of 250 articles were returned from these searches and were filtered according to a developed set of criteria. Criteria used to filter the returned articles included the clear designation of a control group, transparency with methodologies, and relevance to topic. Of the 250 articles initially returned, 40 articles and one database were chosen to be included in this literature review. The articles chosen consisted of 15 animal experimental studies, 11 cell and/or tissue experimental studies, 16 patient observation studies, 6 reviews, and 1 metabolite database, with some overlap between the animal and cell experimental studies. Table 1 lists studies that measured levels of indoxyl sulfate in ASD and other disorders. Table 2 lists studies that conducted in vivo and in vitro cell and animal studies related to the effects of indoxyl sulfate.

## 3. Metabolism of Indoxyl Sulfate

The substance 3-indoxyl sulfate, or indoxyl sulfate, is an aryl sulfate produced by both host and bacterial metabolism in the gut microbiome [6,30]. The molecule consists of a sulfate group bound to an aryl group by an ether as shown in Figure 2 [6]. An amount of 5% of dietary L-tryptophan, acquired through the consumption of protein-rich foods, is metabolized into indole by tryptophanase-expressing bacteria in the gastrointestinal tract (Figure 2) [6,31]. Known gastrointestinal bacteria classifications contributing to the synthesis of indole from L-tryptophan include *Escherichia coli*, *Proteus Vulgaris*, *Clostridium* species, *Bacteroides* species, and *Lactobacillus* species [31,32]. Indole travels through the gastrointestinal tract to the colon, where it is absorbed by the colonic epithelium [31]. In the liver and colonic epithelium, the CYP450 enzyme CYP2E1 processes indole into indoxyl through hydroxylation [6,31,33]. From there, the indoxyl travels to the liver, where it is sulfated into indoxyl sulfate by the SULT1A1 phenol sulfotransferase (PST) enzyme [6]. The fully constructed indoxyl sulfate is then released into the blood, where it binds to proteins such as albumin for transport throughout the body for eventual disposal in the urine [31]. In healthy individuals, 93% of indoxyl sulfate is protein-bound in the blood with the remaining 7% of free indoxyl sulfate excreted in the urine [31]. Indoxyl sulfate may also cross the maternal placenta to an unborn child, with a “moderate fetal-to-maternal perfusate ratio” in healthy term placentas [27].

## 4. Toxin Classification

Indoxyl sulfate is classified as a neurotoxin, uremic toxin, nephrotoxin, cardiotoxin, osteotoxin, and myotoxin [6]. Excess free indoxyl sulfate can aggregate in various body systems and have detrimental effects on the host. An increased concentration of free, non-protein bound indoxyl sulfate can activate aryl hydrocarbon receptors in many cells, increasing the permeability of the blood–brain barrier and inducing cognitive impairment in the host [18]. Additionally, indoles like indoxyl sulfate act as ligands for pregnane X receptors, which aid in the regulation of the integrity of intestinal mucous [34]. An excess of indoles has a direct effect on the properties of the intestinal barrier, and potentially leads to intestinal barrier dysfunction [34]. Indoxyl sulfate may also suppress dynamin-related protein (DRP1) activity through the upregulation of interferon regulatory factor 1 (IRF1), causing mitophagic impairment and intestinal barrier injury [20]. The subsequent sections will examine each toxin classification in detail.

### 4.1. Neurotoxin

Indoxyl sulfate is a known neurotoxin associated with increased permeability of the blood–brain barrier; cognitive impairment; neuropsychiatric issues; changes in an individual’s state of consciousness, conduct, and personality; and impaired neural processing [6,18,22,31]. Indoxyl sulfate’s effects on the blood–brain barrier (BBB) are thought to be among the primary causes of neurological symptoms associated with high concentrations of indoxyl sulfate in the central nervous system, including cognitive dysfunction, poor executive function, and stroke-induced brain injury [12,18]. The disruption of the blood–brain barrier may be caused by indoxyl sulfate’s activation of aryl hydrocarbon receptors (AhR) in central nervous system (CNS) endothelial cells [18].

In cell culture studies, indoxyl sulfate has also been shown to have inflammatory effects on CNS glial cells, with high concentrations of indoxyl sulfate associated with reduced neuronal survival, impaired neural stem cell activity, irregular expression of brain-derived neurotrophic factor and repressor element-1 silencing transcription factor, abnormal serotonin and corticosterone levels, impaired post-receptor intracellular signaling, neuroinflammation, and upregulated oxidative stress [28].

Another cell culture study found that indoxyl sulfate has a variety of effects on different cells involved in the central nervous system. High indoxyl sulfate concentrations increased cellular death in neuronal cell cultures and enhanced nitric oxide (NO), tumor necrosis factor alpha (TNF-α), and interleukin 6 (IL-6) levels in blood sera [17]. These findings were then translated into an in vivo animal study, where the aforementioned compounds induced histological brain alterations and the expression of oxidative stress and inflammatory markers such as cyclooxygenase 2 (COX-2) and nitrotyrosine in the brain [17]. In C6 cells, indoxyl sulfate enhanced reactive oxygen species production, reduced nuclear receptor factor (Nrf2) nuclear translocation; impaired heme oxygenase-1 (HO-1), nicotinamide adenine dinucleotide phosphate (NADPH), quinone dehydrogenase-1 (NQO1), and superoxide dismutase (SOD) expression; and induced AhR activation and p65 nuclear factor kappa beta (NF-κβ) nuclear translocation [17]. Chronic exposure of cultured neurons to indoxyl sulfate impaired neuronal firing rates and induced axonal damage, independent from mitochondrial dysfunction and oxidative stress [25]. Additionally, indoxyl sulfate concentrations were inversely correlated to MRI measurements of cortical volume and directly correlated to levels of neurofilament light chain, which is an established biomarker of neurodegeneration [25].

An animal study performed by Karbowska et al. in 2020 examined the neurobehavioral effects of administration of high doses of indoxyl sulfate on rats [9]. The researchers found that the highest concentrations of indoxyl sulfate in the brain were in the brainstem, which regulates breathing, sleep, blood pressure, heart rate, and consciousness [9]. Chronic elevations of indoxyl sulfate in the brainstem were negatively correlated with locomotor activity, spatial memory, and motor coordination and positively correlated with stress sensitivity and apathetic behavior [9]. Researchers in the study also observed a reduction in norepinephrine, dopamine, and serotonin levels in the brain with an increase in indoxyl sulfate levels [9]. This reduction may help corroborate evidence from other studies showing a positive correlation between indoxyl sulfate concentrations and symptoms of depression, anxiety, neurodegeneration, and other mood disorders [14,28].

A human observational study conducted by Teruya et al. in 2021 performed a comprehensive analysis of blood metabolites in dementia patients using liquid chromatography-mass spectrometry [13]. Researchers from the study found that indoxyl sulfate concentrations in the blood were two times higher in patients with dementia compared to a control group of healthy elderly [13]. The study notes that indoxyl sulfate may have a negative effect on the central nervous system and the brain, thus worsening dementia symptoms in elderly patients [13].

### 4.2. Uremic Toxin

As a uremic toxin, indoxyl sulfate is associated with several related conditions, including chronic kidney disease, glomerular sclerosis, interstitial fibrosis, and renal failure [6]. Indoxyl sulfate concentrations in the urine are elevated in patients with ASD and Parkinson’s disease, indicating a potential correlation between the uremic excretion of the toxin and neurological disorders [1,2]. An amount of 93% of indoxyl sulfate in the body is protein-bound in the blood, leaving the remaining 7% of free indoxyl sulfate to be excreted in the urine [31]. Thus, an increase in urinary indoxyl sulfate may indicate a remarkably higher increase in the overall indoxyl sulfate remaining in the body.

Analogous to indoxyl sulfate, p-cresol sulfate (pCS) is a microbially derived metabolite of phenylalanine instead of tryptophan. Both compounds are uremic toxins derived from essential aromatic amino acids and often co-occur in uremic toxin literature [2]. Further, indoxyl sulfate and pCS follow the same human detoxification pathways and have been discovered to be higher in the urine of ASD populations than in typically developing populations. Together, indoxyl sulfate and p-cresol have been shown to predict mortality among hemodialysis patients [35].

### 4.3. Nephrotoxin

Indoxyl sulfate is also classified as a nephrotoxin and has been associated with chronic kidney disease (CKD), renal tubular cellular necrosis, interstitial fibrosis, renal failure, atherosclerosis, and other forms of kidney disease states [6,36]. In a 2009 study performed by Jourde-Chiche et al., indoxyl sulfate impaired cell antioxidant systems and encouraged pro-inflammatory and pro-fibrotic mechanisms [36]. Indoxyl sulfate also accelerated the progression of chronic renal failure through the increase of transforming growth factor β-1 synthesis in the kidneys, which is known to enhance renal fibrosis [36].

In hemodialysis patients, indoxyl sulfate was associated with aortic calcification, smooth muscle cell proliferation, and endothelial dysfunction [36]. As a toxin primarily bound to proteins in the blood serum, indoxyl sulfate cannot be easily removed from the body via hemodialysis [26]. As a result, the toxin concentrations can increase up to 90 times higher in patients with CKD [26]. Among patients with CKD, only 85% of metabolized indoxyl sulfate is protein-bound in the blood, resulting in a higher percentage of indoxyl sulfate that is free and unbound [31]. This significant increase in free indoxyl sulfate concentrations inhibits the activity of white blood cells—effectively impairing the immune system—and induces inflammation through the enhanced production of inflammatory cytokines in macrophages [26]. Additionally, indoxyl sulfate in high concentrations can cause endoplasmic reticulum stress and epithelial-mesenchymal transition in tubular epithelial cells [37].

A study performed in 2022 examined the effect of indoxyl sulfate on uridine adenosine tetraphosphate-induced contraction in rat renal arteries [24]. The researchers found that a single application of indoxyl sulfate to the isolated Wistar rat renal arteries significantly decreased uridine adenosine tetraphosphate (Up4A), uridine triphosphate (UTP), adenosine triphosphate (ATP), uridine diphosphate (UDP), and adenosine diphosphate (ADP) mediated renal arterial contraction [24]. These results indicate a potential relationship between indoxyl sulfate and vascular function, including the induction of vasoconstriction [24]. The reduction in vascular function may contribute to the development of acute and chronic kidney injuries.

### 4.4. Cardiotoxin

As a known cardiotoxin, indoxyl sulfate can induce endothelial dysfunction, disrupt wound repair, impair angiogenesis, enhance oxidative stress and carbonyl stress, increase oxidative stress via enhanced NADPH oxidase activity in endothelial cells, and reduce levels of glutathione, an active antioxidant system [6]. Additionally, indoxyl sulfate has been associated with several cardiovascular diseases, such as peripheral artery disease and heart rate failure [6,15]. A study performed by Cao et al. in 2015 found an association between the level of indoxyl sulfate plasma concentrations and first heart rate failure events in patients on hemodialysis [15]. The occurrence of first heart rate failure was positively correlated with indoxyl sulfate concentrations, indicating that indoxyl sulfate may potentially be involved in the pathogenesis of the event in a dose-dependent manner [15].

An additional study conducted by Lekawanvijit et al. in 2010 examined the effects of indoxyl sulfate on cultured cardiac myocytes and fibroblasts [23]. The researchers of the study found that indoxyl sulfate directly stimulated cardiac fibroblast collagen synthesis and cardiac myocyte hypertrophy [23]. Indoxyl sulfate was also shown to induce mitogen-activated protein kinase (MAPK) and nuclear factor-κβ (NF-κβ) pathways in cardiac cells and stimulate pro-inflammatory cytokine mRNA expression in human leukemia monocytic (THP-1) cells via the MAPK and NF-κβ pathways [23]. The study further found that indoxyl sulfate could exert these effects on cardiac cells without affecting the cell’s viability, suggesting that indoxyl sulfate may mediate several detrimental effects on cardiac cellular function [23].

Indoxyl sulfate is also connected to instances of heart failure in patients on hemodialysis [15]. A 2015 study followed 258 hemodialysis patients, categorizing their median plasma indoxyl sulfate levels into a “high” group and a “low” group [15]. Patients in the high indoxyl sulfate group were significantly more likely to experience their first heart rate failure event compared to the low indoxyl sulfate group, suggesting that indoxyl sulfate may be involved in the pathogenesis of heart rate failure [15]. However, direct causality cannot be assumed, as poor kidney function may have also been the underlying cause of the patient’s heart failure and elevated indoxyl sulfate levels. More research must be done to make definitive conclusions about indoxyl sulfate’s role as a cardiotoxin in perpetrating heart failure and other cardiac events.

### 4.5. Osteotoxin

Indoxyl sulfate is classified as an osteotoxin due to its recorded influence in abnormal bone turnover and suppressed bone formation and resorption [6]. Indoxyl sulfate may decrease osteoclast differentiation and promote osteoblast apoptosis, resulting in low bone turnover and a reduction in bone quality [38]. At a hormonal level, indoxyl sulfate aggravates vitamin D deficiencies and induces skeletal resistance to parathyroid hormone (PTH) [38,39]. High levels of indoxyl sulfate have been associated with a suppression of PTH-stimulated intracellular cyclic adenosine monophosphate (cAMP) production and a decrease in PTH receptor gene expression and messenger RNA (mRNA) levels [39]. PTH stimulates the release of calcium via osteoclasts, leading to the resorption of bones and greater bone quality [38]. In the context of PTH suppression, abnormal bone formation and resorption can lead to a variety of bone diseases and disorders impacting the patient’s quality of life.

### 4.6. Myotoxin

As a myotoxin, indoxyl sulfate is correlated with the enhanced expression of muscle atrophy-related genes, inflammatory cytokines, and the development of sarcopenia [6]. Additionally, high indoxyl sulfate concentrations have been shown to suppress beta-catenin in microvascular endothelial cells, resulting in an inhibition of wingless-related integration site (Wnt) activity and pro-angiogenic Wnt targets in endothelial cells [40]. The suppression of the Wnt pathway may contribute to the development of adverse limb events such as critical limb ischemia and amputation, particularly in patients with CKD [40].

A study conducted in 2016 by Sato et al. further examined the effects of indoxyl sulfate on skeletal muscle in mice [41]. Researchers from the study found that indoxyl sulfate significantly reduced the cell viability for myoblast cells and myotubules in a dose-dependent manner [41]. Treatment of the cells with indoxyl sulfate increased the cellular metabolites from both the glycolysis and antioxidative response-related pathways and decreased the cellular metabolites of energy generation-related pathways [41]. Furthermore, exposure to indoxyl sulfate significantly increased the intermediate metabolites from the pentose phosphate pathway (PPP) and the glycolysis, glutathione metabolic, and anaerobic metabolic pathways while decreasing the intermediate metabolites in the citric acid (TCA) cycle and glutamate anabolism [41]. Researchers from the Sato et al. study also found a decrease in mitochondrial function and abnormal mitochondrial morphology in muscle cells [41]. An overall increase in plasma indoxyl sulfate levels was associated with skeletal muscle mass reduction and the development of sarcopenia [41].

## 5. Other Disorders with Elevated Levels of Indoxyl Sulfate

Indoxyl sulfate has been associated with a variety of diseases and disorders, including chronic kidney disease, acute kidney injury, Parkinson’s disease, and several mood disorders [7,8,9].

### 5.1. Chronic Kidney Disease & Acute Kidney Injury

As a confirmed nephrotoxin, indoxyl sulfate has been correlated with several kidney conditions and diseases, including chronic kidney disease (CKD) and acute kidney injury (AKI) [15]. In the event of CKD or AKI, the kidneys cannot filter indoxyl sulfate and other toxins out of the body, leading to a build-up of indoxyl sulfate in the kidneys and blood [29]. Because indoxyl sulfate is mostly protein-bound, the toxin is difficult to remove via hemodialysis and therefore tends to build up in patients with CKD [42]. As a result, a strong correlation exists between indoxyl sulfate, more severe CKD symptoms, renal dysfunction, increased blood–brain barrier permeability, poor muscle condition, and a high risk of mortality [19,41,42].

A study conducted by Zhao et al. in 2013 found that indoxyl sulfate concentrations were significantly higher in rats with chronic renal failure compared to their healthy counterparts, in addition to a marked increase in periarteriolar fibrosis, tubulointerstitial fibrosis, and intraglomerular fibrosis in the kidneys [7]. When the researchers administered indoxyl sulfate to the chronic renal failure rats, the severity of renal failure increased, transforming growth factor β-1 expression increased, and tissue inhibitor of metalloproteinase 1 and proα1 collagen increased [7]. These findings were confirmed in an additional study performed on rats, in which indoxyl sulfate was shown to accelerate the progression of chronic renal failure through the increase of transforming growth factor β-1 synthesis in the kidneys [36].

### 5.2. Parkinson’s Disease

Parkinson’s disease has been correlated with high levels of indoxyl sulfate in several patient studies [8,16]. One study conducted in 2015 by Cassani et al. found that patients with Parkinson’s disease had urinary indoxyl sulfate concentrations twice as high as control patients unrelated to constipation, age, gender, or body mass index (BMI) [8]. Another study conducted in 2020 found that Parkinson’s disease patients with motor fluctuations had an almost 1.5 times higher concentration of indoxyl sulfate in the cerebrospinal fluid (CSF) than Parkinson’s patients without motor fluctuations [16]. Additionally, the study found that the ratio of the level of indoxyl sulfate in cerebrospinal fluid to plasma in Parkinson’s patients was four times higher than in the control patient group [16]. This points to a potential relationship between the disease state, the development of Parkinson’s disease, and indoxyl sulfate.

### 5.3. Mood Disorders

Indoxyl sulfate has also been implicated in mood disorders such as anxiety and depression [28]. Indoxyl sulfate urine concentrations are higher among patients with depression compared to their healthy counterparts, with indoxyl sulfate concentrations positively correlated to scores for depression [14]. Such a correlation could be related to indoxyl sulfate’s abnormal influence on neurotransmitters such as norepinephrine, dopamine, and serotonin in the brain [9,28].

A study performed in 2018 by Jaglin et al. demonstrated that high indoxyl sulfate urinary concentrations may have a negative impact on emotional behaviors [21]. Rats with a high production of urinary indoxyl sulfate demonstrated anxiety-like behaviors and vagus nerve activation, suggesting that indoxyl sulfate may influence the brain and behavior [21]. This idea is further supported by indoxyl sulfate’s known effects as a neurotoxin [6]. Another exploratory study performed in 2021 by Brydges et al. found a highly significant positive correlation between indoxyl sulfate serum concentrations and scores for psychic anxiety, total anxiety, and total depression [11]. Indoxyl sulfate was also positively correlated with subcallosal cingulate cortex-functional connectivity (SCC-FC) with the bilateral anterior insula, anterior midcingulate cortex, supplementary motor area, and right premotor area [11]. A positive association between high indoxyl sulfate concentrations and SCC-FC connectivity with the bilateral anterior insula is consistent with the insula’s role in processes related to anxiety, including emotional salience, empathy, and processing uncertainty [11].

## 6. Elevated Levels of Indoxyl Sulfate in Autism

A significant amount of evidence connecting high levels of indoxyl sulfate to ASD has been discussed in the current scientific body of literature [1,2,3,4,5,10]. The association was first noted in 1973 by Mrochek et al., who measured the urinary excretion of indoxyl sulfate in a group of children with ASD [3,43]. Mrochek and his research team found that urinary indoxyl sulfate concentrations were 1.8 times higher in children with severe ASD compared to healthy controls [3]. A study performed by Gevi et al. in 2016 also found that urinary concentrations of indoxyl sulfate were considerably higher in patients with ASD, with concentrations in ASD patients on average being twice that of healthy controls [4]. These findings were corroborated by additional studies performed in 2015, 2020, and 2022, for a total of five studies finding higher levels of urinary indoxyl sulfate concentrations in ASD vs. typically developing children [1,2,3,4,5].

Studies further found a correlation between age and urinary indoxyl sulfate levels, with indoxyl sulfate levels being higher in the urine of children with ASD above age six years compared to healthy controls [1,5]. Children with ASD over age six had an average urinary indoxyl sulfate concentration of 32.63 ± 10.38 μmol/mmol Cr, compared to 18.95 ± 7.11 μmol/mmol Cr in healthy children, nearly a two-fold increase [1]. There was no significant difference in urinary indoxyl sulfate levels in children under the age of six [1].

Indoxyl sulfate urinary concentrations have also been investigated to determine if they relate to severity of symptoms experienced by ASD patients [2,3]. Mrochek et al. noted a positive correlation between urinary indoxyl sulfate levels and the severity of the children’s symptoms in 1973, with concentrations proportionally increasing in relation to a patient’s experience of mild, moderate, or severe symptoms [3]. However, these findings were contradicted in a study performed by Osredkar et al. in 2023, which found that indoxyl sulfate concentrations were highest in the mild ASD group and similar to typically developing children for the moderate and severe ASD groups [2]. Osredkar et al. further found that indoxyl sulfate concentrations did not change significantly based on age or gender, with significance only occurring between patients with ASD and controls [2].

Concentrations of indoxyl sulfate continue to increase in cases with additional diagnoses commonly associated with ASD, such as epilepsy, hearing loss, convulsions, missed childhood milestones, tic disorders, developmental disorders, and attention-deficit hyperactivity disorder [2]. Indoxyl sulfate was not, however, associated with mental regression in children with ASD, nor with a loss of previously acquired skills [10].

Further evidence of indoxyl sulfate’s involvement in ASD is the metabolite’s role in the tryptophan metabolic pathway and the disruption of this pathway in patients with ASD. Of the pathways examined during the study conducted by Gevi et al., the tryptophan metabolic pathway was the most disturbed in ASD patients, with both a high pathway impact and a high statistical significance of *p* > 0.001 [4]. Similar disturbances among multiple urinary indole derivatives and metabolites from the tryptophan metabolic pathway provide substantial evidence of a perturbed tryptophan metabolism in patients with ASD [4]. These disturbances are expected because indoxyl sulfate is derived from tryptophan, so it is likely to affect many aspects of tryptophan metabolism.

## 7. Treatments

Decreasing indoxyl sulfate levels present in the blood may help improve overall patient symptomology by reducing the toxic effects of the metabolite. Noted medicinal treatment options may include Kremezin (AST-120), an oral intestinal absorbent that decreases the serum levels of indoxyl sulfate, typically given to dialysis patients to improve their prognosis [36]. However, there are concerns that it may also absorb nutrients and other medications or supplements. Additionally, dietary treatments may be an option for naturally reducing the amount of indoxyl sulfate produced by gut bacteria. These dietary treatment options may include reducing the number of dairy products and protein-rich foods consumed by the patient and increasing the consumption of fruits and vegetables [8,14,42]. A portion of bodily tryptophan—a precursor for indoxyl sulfate—is acquired by ingesting protein-rich foods [31]. By decreasing the consumption of such foods, patients can reduce their intake of tryptophan, thus reducing their overall tryptophan metabolism and the production of indole derivatives, such as indoxyl sulfate [42]. Another option may be microbiota transplant therapy, the transplant of microbiota from a healthy person to a person with autism [44].

## 8. Discussion

Indoxyl sulfate was elevated in six out of six articles examining indoxyl sulfate levels in patients with ASD, with indoxyl sulfate levels averaging twice that of healthy control patients [1,2,3,4,5,10]. Indoxyl sulfate has many toxic effects, including acting as a neurotoxin, uremic toxin, nephrotoxin, cardiotoxin, osteotoxin, and myotoxin. Indoxyl sulfate is also associated with numerous comorbidities associated with ASD, including chronic kidney disease, Parkinson’s disease, and mood disorders such as depression and anxiety [14,42,45]. This indicates that indoxyl sulfate may be involved in the development of these comorbid conditions, resulting in the patient experiencing more intense symptoms. Additional research should be conducted to determine if the reduction of indoxyl sulfate concentrations using various treatment methodologies reduces the symptoms of ASD experienced by the patient. Confirming this would result in a greater understanding of indoxyl sulfate’s influence on ASD symptoms and more definitive answers regarding the cause of said symptoms.

Indoxyl sulfate may also influence ASD in a more indirect way. For instance, indoxyl is sulfated by the body into indoxyl sulfate to prepare the metabolite for excretion in the urine, thus reducing the overall sulfate pool [31]. Children with autism have limited sulfation capacity, so a further reduction in the sulfate pool will decrease the ability to sulfate other toxins and metabolites and may also contribute to increased symptom severity [46].

Additionally, there is a lack of understanding regarding the relative toxicity of albumin-bound indoxyl sulfate compared to free indoxyl sulfate. Little is known of the bound form’s toxicity and if bound indoxyl sulfate carries out similar effects on the body to free indoxyl sulfate. Additional research is needed to explore these gaps in the current understanding of indoxyl sulfate and its toxic effects on the body.

Due to indoxyl sulfate’s elevated levels in ASD, the metabolite could serve as a potential biomarker for a subset of ASD patients and as a biomarker for one type of intestinal dysbiosis.

## 9. Limitations

Most cell and animal experimental studies administered very high doses of indoxyl sulfate to see conclusive results. It is difficult to make conclusions about the effects of lower chronic dosages on humans, including children with ASD. Additional research is necessary to explore the effect of free indoxyl sulfate versus albumin-bound indoxyl sulfate, which occurs in a much higher concentration in the blood. Measuring the relative amount of each present within the body would provide additional information about the source of potential symptoms experienced by the patient and garner insight into each of the two forms’ (protein-bound and free) toxicities.

## 10. Conclusions

Indoxyl sulfate is a neurotoxin, a uremic toxin, a nephrotoxin, a cardiotoxin, an osteotoxin, and a myotoxin [6]. Elevated levels of indoxyl sulfate are associated with several conditions, including chronic kidney disease, acute kidney injury, Parkinson’s disease, and various mood disorders including anxiety and depression [7,8,9]. Indoxyl sulfate has been found to be higher in children with ASD compared to typically developing children in six out of six studies [1,2,3,4,5,10]. Adults with ASD have much higher rates of chronic kidney disease, Parkinson’s disease, and mood disorders, further suggesting that indoxyl sulfate may also contribute to co-morbid symptoms in ASD. The elevated levels of indoxyl sulfate in patients with ASD and its known toxic effects suggest that it contributes to ASD symptoms and that abnormal gut bacteria are important to the etiology of autism. The current research presented in this literature review suggests indoxyl sulfate’s potential as a biomarker for a subset of ASD patients and strong potential as a biomarker for intestinal dysbiosis.

## Figures and Tables

**Figure 1 ijms-25-12973-f001:**
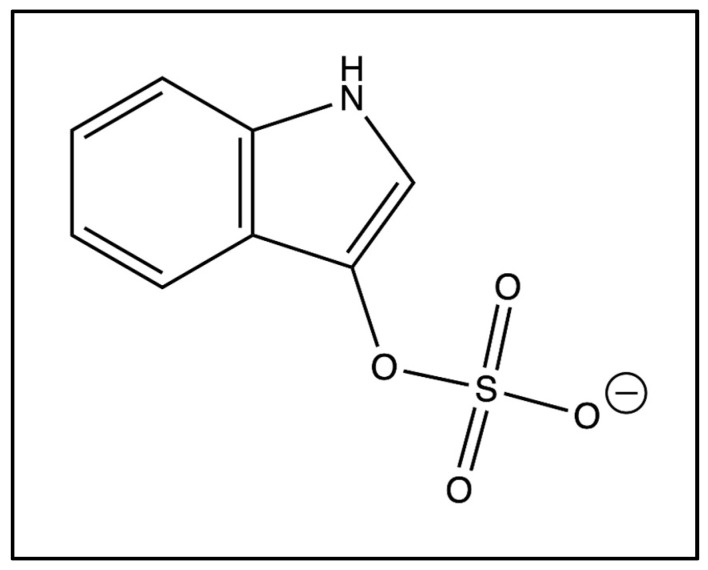
Chemical structure of indoxyl sulfate.

**Figure 2 ijms-25-12973-f002:**
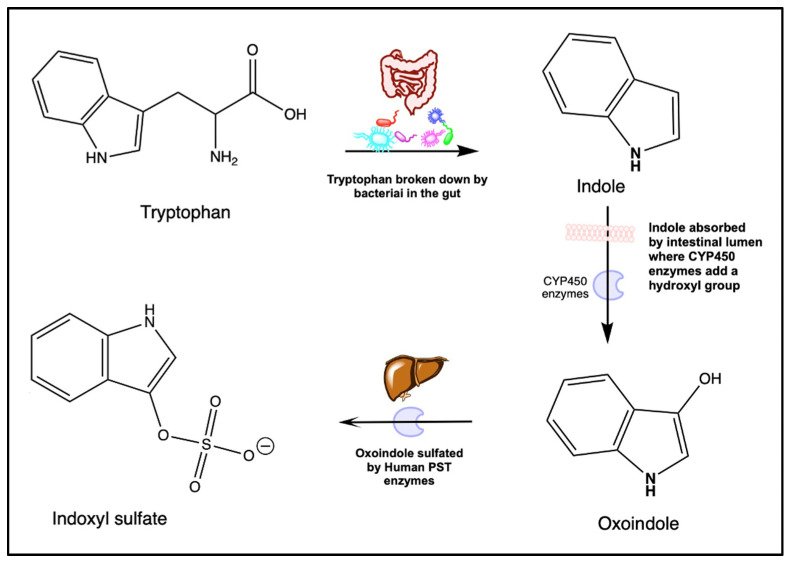
Indoxyl sulfate synthesis in human metabolic systems.

**Table 1 ijms-25-12973-t001:** Studies that measured levels of indoxyl sulfate in ASD and other disorders.

Author	Condition	Type of Sample	Patient Levels	Control Levels	Patient/Control Ratio	Findings
Dieme 2015 [5]	Autism	Urine	Visual Data	Visual Data	Not applicable	IS urinary concentrations higher in ASD patients compared to controls (*p =* 0.01)
Gevi 2016 [4]	Autism	Urine	Not applicable	Not applicable	Not applicable	IS urinary concentrations higher in ASD patients compared to controls (*p <* 0.001)
Mrochek 1973 [3]	Autism	Urine	Mild: 4 mg/24 h, Severe: 6 mg/24 h, Neurological Seizures: 7 mg/24 h	3.4 mg/24 h	Mild: 1.18, Severe: 1.76, Seizures: 2.06	IS urinary concentrations higher in ASD patients compared to controls; concentrations increase with severity
Olesova 2020 [1]	Autism	Urine	32.63 ± 10.38 μmol/mmol Creatinine (Cr)	18.95 ± 7.11 μmol/mmol Cr	1.72	IS urinary concentrations higher in ASD patients compared to controls (*p =* 0.00004)
Osredkar 2023 [2]	Autism	Urine	55.64 μg/mg	52.62 μg/mg	1.06	IS urinary concentrations higher in ASD patients compared to controls (*p =* 0.0182)
Rangel-Huerta 2019 [10]	Autism	Plasma	Visual Data	Visual Data	Not applicable	IS urinary concentrations higher in ASD patients with no regression compared to controls (*p <* 0.001)
Brydges 2021 [11]	Anxiety	Serum	Visual Data	Visual Data	Not applicable	IS significantly correlated with Hamilton Psychic Anxiety (*p =* 0.0001) and Hamilton Total Anxiety (*p =* 0.002)
Yeh 2016 [12]	Cognitive Impairment	Serum	22.0 ± 25.8 mg/mL	4.8 ± 5.1 mg/mL	4.58	IS independently associated with poor executive function; correlation varried depending on stage of Chronic Kidney Disease (CKD)
Teruya 2021 [13]	Dementia	Plasma	Not applicable	Not applicable	1.93	IS higher in dementia patients’ plasma than in healthy elderly (*p =* 0.015)
Philippe 2021 [14]	Depression	Urine	234.2 μmol/mg Cr	197.3 μmol/mg Cr	1.19	IS urinary concentrations higher in ASD patients compared to controls (*p =* 0.0098)
Cao 2015 [15]	Heart Failure	Plasma	Low IS Plasma Group: £32.35 mg/mL High IS Plasma Group: >32.35 mg/mL	Not applicable	Not applicable	Incidence of first heart failure event significantly higher in high IS group compared to low IS group (*p <* 0.001)
Cassani 2015 [8]	Parkinson’s	Urine	48.3 ± 4.9 mg/L	24.9 ± 4.6 mg/L	1.94	Parkinson’s patients had urinary IS concentrations twice as high as controls (*p =* 0.001)
Sankowski 2020 [16]	Parkinson’s	Cerebrospinal fluid (CSF)/Plasma	CSF/Plasma: 1.6 ± 0.9%	CSF/Plasma: 0.42 ± 0.21%	3.81	IS concentrations significantly higher in Parkinson’s cerebrospinal fluid (CSF) with motor fluctuation compared to Parkinson’s without motor fluctuation (*p <* 0.05); CSF to plasma ratio of IS in Parkinson’s group four times greater than control (*p <* 0.05)

**Table 2 ijms-25-12973-t002:** Studies that conducted in vivo and in vitro cell and animal studies related to the effects of indoxyl sulfate.

Author	Indoxyl Sulfate Administered	Groups	R-Value & *p*-Value
Adesso 2017 [17]	Cells: 15–60 mM for 24 h; Injected mice: 800 mg/kg IS	C6 cells group; Primary astrocytes and mixed glial cells group; Injected mice group; Control mice group	Enhanced reactive oxygen species (ROS) in C6 cells: *p <* 0.001; Reduced nuclear factor erythroid 2-related factor 2 (Nrf2) Nuclear Translocation in C6 cells: *p <* 0.01; Reduced heme oxygenase 1 (HO-1), nicotinamide adenine dinucleotide phosphate (NADPH) quinone dehydrogenase 1 (NQO1), and superoxide dismutase (SOD) expression in C6 cells: *p <* 0.01; Induced aryl hydrocarbon receptor (AhR) activation in C6 cells: *p <* 0.001; Induced p65 nuclear factor kappa-B (NF-kB) nuclear translocation in C6 cells: *p <* 0.05; Influenced oxidative stress and pro-inflammatory parameters in primary astrocytes and mixed glial cell cultures: *p <* 0.001 (greater effect on mixed culture than astrocytes alone: *p <* 0.05); Increased cellular death in neuronal cultures: *p <* 0.05; IS serum concentration compared to control: *p <* 0.001
Bobot 2020 [18]	1 g/L of water for 14 days	Rat Group 1: Induced Chronic Kidney Disease (CKD) (adenine-rich diet or 5/6 nephrectomy) Rat Group 2: Control	Blood brain barrier (BBB) permeability & Cognitive Impairment: R = −0.90, *p <* 0.001; BBB permeability & IS levels: R = 0.68, *p =* 0.006
Griffin 2023 [19]	100 mg/kg body weight, 200 mg/kg body weight	Pregnancy-related acute kidney injury (AKI) group of rats; Normal pregnant 100 mg/kg group; Normal pregnant 200 mg/kg; Normal pregnant group	Severity of decrease in weight associated with amount of IS administered: *p =* 0.002; IS administration significantly increased BBB permeability: *p <* 0.0001
Huang 2020 [20]	IS-injected mice (intraperitoneal injection, 100 mg/kg daily for 8 weeks)	Transepithelial electrical resistance, permeability assay and transmission electron microscopy carried out to evaluate the damaging effect of IS on intestinal barrier in intestinal epithelial cells (in vitro study); IS-injected mice (intraperitoneal injection, 100 mg/kg daily for 8 weeks), and CKD mice (in vivo studies); CKD mice then treated with Kremezin (AST-120) (oral absorbent for IS) and gene knockout mice used to verify mechanism and explore possible interventions for IS-induced intestinal barrier injury	Not applicable
Jaglin 2018 [21]	Injected 500 mg/kg in rat cecum, also colonized germ-free rats with indole producing bacteria *E. coli*	Injection experiment: 12 injected, 12 control (injected with corn oil), Colonization experiment: 12 I+, 12 I− (control)	Injection experiment: distance traveled (*p <* 0.001), rearing (*p <* 0.001), eye blinking frequency (*p <* 0.001), Colonization experiment: Eye blinking frequency (*p <* 0.01)
Karbowska 2020 [9]	100–200 mg/kg body weight/day for 4 weeks	Rat Group 1: 100 mg/kg body weight/day; Rat Group 2: 200 mg/kg body weight/day; Rat Group 3: Control	IS in Brainstem: control vs. 100 IS: *p =* 0.0093; control vs. 200 IS: *p <* 0.0001; Groomings: *p =* 0.0371; Time to exit: *p =* 0.0133; Latency: *p =* 0.0002; Correct Response: *p =* 0.0116
Lai 2022 [22]	Profile of fecal, blood sera, and cerebral cortical brain tissues	Germ-free C57BL/6 mice and their age-matched conventionally raised specific-pathogen-free counterparts	Not applicable
Lekawanvijit 2010 [23]	Isolated rat cardiac myocytes and fibroblasts and human leukaemia monocytic cell line cells incubated with 0.1 nM to 300 mM	Control group; IS incubation groups	Neonatal cardiac fibroblasts (NCF) collagen synthesis: *p <* 0.05; Neonatal cardiac myocytes (NCM) hypertrophy: *p <* 0.001
Matsumoto 2022 [24]	Isolated rat renal arteries incubated in 1.0 mM indoxyl sulfate for 1 h	Control group; IS incubation group	*p <* 0.05
Ntranos 2022 [25]	Cultured rat neurons exposed to cerebrospinal fluid (CSF) samples collected from multiple sclerosis patients	20 healthy controls; 16 relapsing remitting multiple sclerosis patients (RRMS); 17 patients with secondary progressive multiple sclerosis (SPMS)	Levels of IS in RRMS patients compared to control: *p =* 0.0537; Levels of IS in SPMS patients compared to control: *p =* 0.03795; Correlation of IS levels to cortical volume: r = −0.45, *p =* 0.037; Correlation of IS levels to levels of neurofilament light chain: r = 0.58, *p =* 0.011
Ribeiro 2023 [26]	250 mM (low concentrations), 60 mg/mL media (high concentrations) checked at 24 and 48 h	Macrophages and tubular epithelial (UT7/EPO) cells, incubated with IS at low concentrations and concentrations found in uremic patients, alone and with lipopolysaccharide challenge; control group	*p <* 0.05
Schakenraad 2021 [27]	Not applicable	Placental transfer studied in healthy term placentas via ex vivo dual-side human cotyledon perfusion technique	Not applicable
Sun 2021 [28]	100 mg/kg body weight	Rat Group 1: Control; Rat Group 2: Unilateral nephrectomy & 100 mg/kg	*p <* 0.05
Won 2016 [29]	Not applicable	Cisplatin-induced acute kidney injury (AKI) rats; Control group	*p <* 0.01
Zhao 2012 [7]	Not applicable	Rat Group 1: Adenine-induced chronic renal failure; Rat Group 2: Control	*p <* 0.001

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
