# Peer review of "Indoxyl Sulfate and Autism Spectrum Disorder: A Literature Review"

_ijms, 2024, doi:10.3390/ijms252312973_

Round 1
Reviewer 1 Report
Comments and Suggestions for Authors
Manuscript entitled „ Indoxyl Sulfate and Autism Spectrum Disorder: A Literature Review” is very interesting, well-written and well-planned review article. I fully support the publication of this manuscript; however I recommend the minor revision of manuscript. Some corrections should be made to the text according to the following comments:
Methods
Table 1 and 2 - in this form the tables are unreadable. All abbreviations used in the tables for the first time must be explained by their full name
Metabolism of Indoxyl Sulfate
Line 90 – write Figure 1
Figure 2 - All abbreviations used in the figures for the first time must be explained by their full name
Toxin Classification
Line 130 – explain in full name an abbreviation CNS
Line 132 -use only an abbreviation CNS, remove full name
Line 146 - explain in full name an abbreviation NADPH
Line 213 - explain in full name abbreviations Up4A-, UTP-, ATP-, UDP-, and ADP
Line 257 - explain in full name an abbreviation cAMP
Line 258 - explain in full name an abbreviation mRNA
Line 267 - explain in full name an abbreviation WNT
Other Disorders with Elevated Levels of Indoxyl Sulfate
Line 314 - explain in full name an abbreviation BMI
Elevated Levels of Indoxyl Sulfate in Autism
Line 371 - explain in full name an abbreviation TD
Treatments
Line 391 - explain in full name an abbreviation AST-120
References
Check all the references and correct them according to the style preferred by the journal
Journal Articles:
1. Author 1, A.B.; Author 2, C.D. Title of the article. Abbreviated Journal Name Year, Volume, page range.
- no citation of reference number 6 in the text of the manuscript
- after reference number 20, reference 22 is cited, but it should be reference 21.
This is an error that should be corrected. Individual literature items must be cited in the text of the manuscript in numerical order.

Author Response
Comment 1: Table 1 and 2 - in this form the tables are unreadable. All abbreviations used in the tables for the first time must be explained by their full nameResponse 1: Thank you for pointing this out. We agree with this comment. Therefore, we have expanded all our abbreviations in tables 1 and 2.
Comment 2: Line 90 – write Figure 1
Response 2: Thank you for pointing this out. We agree with this comment. Therefore, we have written Figure 2 on Line 90.
Comment 3: Figure 2 - All abbreviations used in the figures for the first time must be explained by their full name
Response 3: Thank you for pointing this out. We agree with this comment. Therefore, we have expanded all our abbreviations from figure 2 in the corresponding text.
Comment 4: Line 130 – explain in full name an abbreviation CNS
Response 4: Thank you for pointing this out. We agree with this comment. Therefore, we have expanded “CNS” to “central nervous system (CNS)” on line 130.
Comment 5: Line 132 -use only an abbreviation CNS, remove full name
Response 5: Thank you for pointing this out. We agree with this comment. Therefore, we have abbreviated “central nervous system” to “CNS” on line 130.
Comment 6: Line 146 - explain in full name an abbreviation NADPH
Response 6: Thank you for pointing this out. We agree with this comment. Therefore, we have expanded “NADPH” to “nicotinamide adenine dinucleotide phosphate (NADPH)” on line 146.
Comment 7: Line 213 - explain in full name abbreviations Up4A-, UTP-, ATP-, UDP-, and ADP
Response 7: Thank you for pointing this out. We agree with this comment. Therefore, we have expanded “Up4A-, UTP-, ATP-, UDP-, and ADP” to “uridine adenosine tetraphosphate (Up4A), uridine triphosphate(UTP), adenosine triphosphate (ATP), uridine diphosphate (UDP), and adenosine diphosphate (ADP) uridine adenosine tetraphosphate (Up4A), uridine triphosphate (UTP), adenosine triphosphate (ATP), uridine diphosphate (UDP), and adenosine diphosphate (ADP)” on lines 215-217.
Comment 8: Line 257 - explain in full name an abbreviation cAMP
Response 8: Thank you for pointing this out. We agree with this comment. Therefore, we have expanded “cAMP” to cyclic adenosine monophosphate (cAMP) on line 261.
Comment 9: Line 258 - explain in full name an abbreviation mRNA
Response 9: Thank you for pointing this out. We agree with this comment. Therefore, we have expanded “mRNA” to “messenger RNA” on line 262.
Comment 10: Line 267 - explain in full name an abbreviation WNT
Response 10: Thank you for pointing this out. We agree with this comment. Therefore, we have expanded “WNT” to “wingless-related integration site (Wnt) wingless-related integration site (Wnt)” on lines 272-273.
Comment 11: Line 314 - explain in full name an abbreviation BMI
Response 11: Thank you for pointing this out. We agree with this comment. Therefore, we have expanded “BMI” to “body mass index (BMI)” on line 320.
Comment 12: Line 371 - explain in full name an abbreviation TD
Response 12: Thank you for pointing this out. We agree with this comment. Therefore, we have expanded “TD” to “typically developing children” on line 377.
Comment 13: Line 391 - explain in full name an abbreviation AST-120
Response 13: Thank you for pointing this out. We agree with this comment. Therefore, we have expanded “AST-120” to “Kremezin (AST-120)” on line 397.
Comment 14: Check all the references and correct them according to the style preferred by the journal
Journal Articles: Author 1, A.B.; Author 2, C.D. Title of the article. Abbreviated Journal NameYear, Volume, page range.
Response 14: Thank you for pointing this out. We agree with this comment. Therefore, we have fixed the references to reflect this suggestion beginning on line 471.
Comment 15: No citation of reference number 6 in the text of the manuscript
Response 15: Thank you for pointing this out. We agree with this comment. Therefore, we have fixed the references within the manuscript to reflect this suggestion.
Comment 16: After reference number 20, reference 22 is cited, but it should be reference 21. This is an error that should be corrected. Individual literature items must be cited in the text of the manuscript in numerical order.
Response 16: Thank you for pointing this out. We agree with this comment. Therefore, we have fixed the references within the manuscript to reflect this suggestion.
Reviewer 2 Report
Comments and Suggestions for Authors
Zoë R. Hill et al present a review of the literature around the association of increased levels of Indoxyl Sulfate in several pathologies, including Autism Spectrum Disorders. The topic of the manuscript is of interest and also reflects the gaps of knowledge at present about this topic. The table and figures are clear and summarize the information presented. The content of the manuscript is well structured and clearly presents the information in a meaningful way to the reader. The literature reviewed is up to date and the manuscript follows a logical order, which makes it easy to read. Overall, the manuscript is very detailed, well written and summarizes the current state of play.
I just have two minor points, which are detailed below and should be addressed before publication.
1- lines 41-2: “Evidence linking the gut microbiome and gastrointestinal dysfunction to ASD has increased in recent years [2].” Please, include more references in order to support your statement.
2- lines 13-4 and 88-9: Authors stated “Indoxyl sulfate is derived from bacterial modification of tryptophan”, and also “3-indoxyl sulfate, or indoxyl sulfate, is an aryl sulfate produced by both host and bacterial metabolism in the gut microbiome [3], [8]”. Please, clarify the apparent contradiction.
3- lines 182-188: “Analogous to indoxyl sulfate, p-cresol sulfate (pCS) is a microbially-derived metabolite of phenylalanine instead of tryptophan. Both compounds are uremic toxins derived from essential aromatic amino acids, and often co-occur in uremic toxin literature [2]. Further, indoxyl sulfate and pCS follow the same human detoxification pathways and have been discovered to be higher in the urine of ASD populations than in typically developing populations. Together, indoxyl sulfate and p-cresol have been shown to predict mortality among hemodialysis patients [23].” I wonder if there is difference between pCS and indoxyl sulfate in terms of association with ASD? In other words, was PCS also associated with ASD? This paragraph suggests that the both metabolites have similar toxic effects, thus the logical question that arises is why did you focus only on indoxyl sulfate?
4- Line 268: Please, define WNT.
5- Line 371: Please, define TD.
6- Lines 429-431: “ Due to indoxyl sulfate’s elevated levels in ASD, the metabolite could serve as a potential biomarker for a subset of ASD patients, and as a biomarker for one type of intestinal dysbiosis.” Please, clear up what “a subset of ASD patients” means.
Author Response
Comment 1: lines 41-2: “Evidence linking the gut microbiome and gastrointestinal dysfunction to ASD has increased in recent years [2].” Please, include more references in order to support your statement.Response 1: Thank you for pointing this out. We agree with this comment. Therefore, we have supplemented the references for this statement on line 42.
Comment 2: lines 13-4 and 88-9: Authors stated “Indoxyl sulfate is derived from bacterial modification of tryptophan”, and also “3-indoxyl sulfate, or indoxyl sulfate, is an aryl sulfate produced by both host and bacterial metabolism in the gut microbiome [3], [8]”. Please, clarify the apparent contradiction.
Response 2: Thank you for pointing this out. We agree with this comment. Therefore, we have changed the wording of the first statement to “Indoxyl sulfate is derived from bacterial modification of host tryptophan” on lines 13-14.
Comment 3: lines 182-188: “Analogous to indoxyl sulfate, p-cresol sulfate (pCS) is a microbially-derived metabolite of phenylalanine instead of tryptophan. Both compounds are uremic toxins derived from essential aromatic amino acids, and often co-occur in uremic toxin literature [2]. Further, indoxyl sulfate and pCS follow the same human detoxification pathways and have been discovered to be higher in the urine of ASD populations than in typically developing populations. Together, indoxyl sulfate and p-cresol have been shown to predict mortality among hemodialysis patients [23].” I wonder if there is difference between pCS and indoxyl sulfate in terms of association with ASD? In other words, was PCS also associated with ASD? This paragraph suggests that both metabolites have similar toxic effects, thus the logical question that arises is why did you focus only on indoxyl sulfate?
Response 3: Thank you for pointing this out. This is a very good question. There is a lot of information on p-cresol sulfate, which we are covering in a separate paper. We think it would be less confusing and more impactful to present the two compounds in two separate manuscripts.
Comment 4: Line 268: Please, define WNT.
Response 4: Thank you for pointing this out. We agree with this comment. Therefore, we have expanded “WNT” to “wingless-related integration site (Wnt) wingless-related integration site (Wnt)” on lines 272-273.
Comment 5: Line 371: Please, define TD.
Response 5: Thank you for pointing this out. We agree with this comment. Therefore, we have expanded “TD” to “typically developing children” on line 377.
Comment 6: Lines 429-431: “Due to indoxyl sulfate’s elevated levels in ASD, the metabolite could serve as a potential biomarker for a subset of ASD patients, and as a biomarker for one type of intestinal dysbiosis.” Please, clear up what “a subset of ASD patients” means.
Response 6: Thank you for pointing this out. By this statement, we mean that a subset of patients with ASD have high levels of IS, not all.